



# Latitudinal distribution of biomarkers across the western Arctic Ocean and the Bering Sea: an approach to assess sympagic and pelagic algal production

5    Youcheng Bai[1,2], Marie-Alexandrine Sicre[3], Jian Ren[1,2], Vincent Klein[3], Haiyan Jin[1,2,4] and Jianfang Chen[1,2,5*]

[1]Key Laboratory of Marine Ecosystem Dynamics, Ministry of Natural Resources, Hangzhou 310012, China

10   [2]Second Institute of Oceanography, Ministry of Natural Resources, Hangzhou 310012, China

[3]LOCEAN, CNRS, Sorbonne Université, Campus Pierre et Marie Curie, Case 100, 4 Place Jussieu, 75032, Paris, France

[4]School of Oceanography, Shanghai Jiao Tong University, Shanghai 200230, China

[5]State Key Laboratory of Satellite Ocean Environment Dynamics, Second Institute of

15   Oceanography, Ministry of Natural Resources, Hangzhou 310012, China

*Corresponding to*: Jianfang Chen (jfchen@sio.org.cn)



**Abstract.** The drastic decline of Arctic sea ice due to global warming and polar amplification of environmental changes in the Arctic basin profoundly alter primary production with consequences for polar ecosystems and the carbon cycle. In this study, we use highly branched isoprenoids (HBIs), brassicasterol, dinosterol and terrestrial biomarkers (*n*-alkanes and campesterol) in surface sediments to assess sympagic and pelagic algal production with changing sea ice conditions along a latitudinal transect from the Bering Sea to the high latitudes of the western Arctic Ocean. Suspended particulate matter (SPM) was also collected in surface waters at several stations of the Chukchi basin to provide snapshots of phytoplankton communities under various sea ice conditions for comparison with underlying surface sediments. Our results show that sympagic production (IP$_{25}$ and HBI-II) increased northward between 62°N and 73°N, with maximum values at the sea ice edge in the Marginal Ice Zone (MIZ) between 70°N and 73°N in southeastern Chukchi Sea and along the coast of Alaska. They were consistently low at northern high latitudes (>73°N) under perennial sea ice and in the Ice-Free Zone (IFZ) of the Bering Sea. Enhanced pelagic sterols and HBI-III occurred in the IFZ across the Bering Sea and in southeastern Chukchi Sea up to 70°N-73°N in the MIZ conditions that marks a shift of sympagic over pelagic production. In surface water SPM, pelagic sterols display similar patterns as Chl *a,* increasing southwards with higher amounts found in the Chukchi shelf pointing out the dominance of diatom production. Higher cholesterol values were found in the mid-Chukchi Sea shelf where phytosterols were also abundant. This compound prevailed over phytosterols in sediments, compared to SPM, reflecting efficient consumption of algal material in the water column by herbivorous zooplankton.



## 1. Introduction

The Arctic Ocean is undergoing the most rapid climate and environmental changes of the world ocean with notably the drastic reduction of sea ice cover and thickness and increased freshwater from higher Arctic river flows (IPCC, 2021). This rapid sea-ice retreat due to global warming has major consequences not only on sea ice ecosystems thriving in and below sea ice and their diversity, but also on pelagic phytoplankton and zooplankton communities and

subsequently on the food chain (Ardyna et al., 2014; Ardyna and Arrigo, 2020). Surface freshening caused by sea ice melting and higher river run-off induce changes of phytoplanktonic species (Park et al., 2023) thereby altering the composition, export and sequestration of organic carbon in marine sediments and ultimately the Arctic marine carbon sink (Coupel et al., 2015; Brown, et al., 2020; Su et al, 2023). In general, at the annual bloom

beginning period (April-July), diatoms and large phytoplankton are still dominant, while the percentage of smaller phytoplankton increases later summer/early autumn, and thus organic carbon fluxes to the sediments is mainly driven by larger phytoplankton cells (Moran et al., 2012). Li et al. (2009) evidenced a shift towards smallest phytoplankton cells at the expense of larger cells as a result of upper water column stratification. While smaller size picoplankton is

efficient in using nutrients and light under stratified conditions, it is less prone to sinking due to low cell density and thus does not support large vertical export as opposed to micro phytoplankton such as diatoms (Li et al., 2009). Such shifts in the quantity and phytoplankton communities need to be better assessed to understand production pathways and impact on secondary production, including microbial communities, and higher trophic levels as well as

export and sequestration of organic carbon to the deep ocean, known as the biological pump (Lannuzel et al., 2020).

 Sea ice provides a habitat for a wide array of microalgae, bacteria, autotrophic, mix trophic and heterotrophic protists (Gradinger, 1999; Hop et al., 2020). Ice-associated production (sympagic) is higher in the bottom of the sea ice due to the nutrient supply from the underlying

seawater (Brown et al., 2011; Arrigo, 2017). Ice algae are also an important component in the transfer of organic carbon to deeper layers because they form aggregates that can sink faster (Boetius et al., 2013). Usually, they sink very fast as episodic events similar to sea ice melting. Estimates of the contribution of sympagic to primary production varies from 0 to 80% depending on the sea ice type, region and season and are thus difficult to assess basin-wide

(Legendre et al., 1992; Gosselin et al., 1997; Ehrlich et al., 2021). Boetius et al. (2013) reported that in summer 2012 of unprecedented decline of sea ice, macro-aggregates of diatom *Melosira arctica* contributed at least 45% of total primary productivity and more than 85% of carbon export in the central Arctic basin between 82°N and 89°N. The increased area of thin seasonal sea ice and melt ponds with sea ice due to global warming is likely to affect phytoplankton



carbon uptake and export. The longer season for primary production due to earlier melting and later freezing is another parameter that will likely enhance carbon uptake in the future.

Diagnostic biomarkers can be used to evaluate sympagic and pelagic primary production. Indeed, highly branched isoprenoids (HBIs), with $C_{20}$, $C_{25}$ and $C_{30}$ hydrocarbon are formed by an unusual linkage of C5 isoprene units unique to marine diatoms (Volkman et al., 1994; Belt

et al., 2007). They are only selectively biosynthesised by some diatoms including *Haslea, Pleurosigma, Navicula and Rhizosolenia*, and limited to a small number of species within these taxa (Brown et al., 2014a; Belt et al., 2017). The mono-unsaturated $C_{25}$-HBIs (Ice Proxy with 25 carbon atoms or $IP_{25}$) was initially proposed by Belt et al. (2007) as a proxy of seasonal sea ice. Since the first high-resolution reconstruction published by Massé et al. (2008) in the Nordic

Seas, this biomarker has been widely used in paleoceanographic studies often combined with other HBIs or phyto-sterols to achieve semi-quantitative estimates of  seasonal sea ice extent (Belt, 2018; Kolling et al., 2020).  Co-occurring with $IP_{25}$ in the Arctic Ocean, HBI-II is also sea-ice related and proven to be useful as an additional sea ice proxy (Belt and Müller, 2013). In recent studies, the HBI-III has been proposed as reflecting pelagic production, partly owing

to its sediment distribution in ice-edge or ice-free conditions (Belt et al., 2015; Smik et al., 2016; Schmidt et al., 2018).

This study in the western Arctic Ocean aims at characterizing the distribution of sedimentary HBIs and selected sterols across a latitudinal transect from 54.6 °N to 85.4 °N from the Bering Sea to the Chukchi Sea and explore the relationship between sympagic-pelagic algal production

and seasonal sea ice. Suspended particles were also collected in surface waters in summer 2014 at 13 stations along the same transect for sterol analyses and compared with surface sediments (proximity to the coast, ice cover, etc…) to investigate export pathways.

## 2.  Regional settings

The two main surface ocean currents of the Bering Sea basin are the eastward Aleutian North Slope Current (ANSC) flowing along the Aleutian Islands (Stabeno and Reed, 1994) and the Bering Slope Current (BSC) flowing along the Bering Sea Slope, originating from the inflow of Alaskan Stream (AS) and driving a large-scale cyclonic circulation in the Bering Sea (Fig. 1). The main hydrological features of the Bering Sea northern shelf include the Anadyr Water

(AW) originating from the BSC and the Alaska Coastal Water (ACW) while the surface hydrology of the Chukchi Sea is strongly influenced by the warm northward flowing Pacific Water (PW) entering the Arctic Ocean through the Bering Strait and the seasonal cover of sea ice both playing an important role in the ecology and phytoplankton structure. The PW has an annual mean transport rate of 0.8 Sv (Roach et al., 1995; Woodgate et al., 2005) and comprises

three water masses (Coachman et al., 1975): i) the colder, saline (> 32.5) and nutrient-rich AW



(Grebmeier et al., 1988; Weingartner et al., 2005) in the western side of the northern Bering and Chukchi Sea; ii) the relative warm, low-salinity(< 31.8) and nutrient-depleted ACW (Woodgate and Aagaard, 2005; Hunt et al., 2013) in the eastern side and in between the Bering Shelf Water (BSW) with moderate salinity (31.8-32.5; Woodgate et al., 2005) and nutrient

levels, iii) the seasonal southward-flowing Siberian Coastal Current (SCC) in the western Chukchi Sea that usually deflects small amounts of cold fresh waters (~0.1Sv) into the central Chukchi Sea (Weingartner et al., 1999) (Fig. 1).

The northern Bering-Chukchi Sea is one of the largest marginal seas in the Arctic Ocean (Jakobsson et al., 2014) and one of the most productive areas (Arrigo & van Dijken, 2011; Hill

et al., 2018). Because of continued nutrients supply from the PW, surface waters in this region maintain a high primary production both during the sea ice melting season and ice-free waters in summer. With the retreat of sea ice, the algal production associated with sea ice (Gradinger, 2009) as well as the under-ice phytoplankton blooms (Arrigo et al., 2012; Coupel et al., 2012, 2015; Ardyna et al., 2020) provide food to higher trophic levels (Ji et al., 2012; Kohlbach et al.,

2016; Tedesco et al., 2019; Cautain et al., 2022). With the thinning and retreat of sea ice triggered by global warming, production and export of carbon from sea-ice algal in the Arctic Ocean will also become increasingly important (Ardyna and Arrigo, 2020). However, the exact implications of the drastic sea ice reduction of the last decades on marine ecosystems are difficult to predict because of adverse effects on phytoplankton (Shimada et al., 2006; Harada

et al., 2016).

## 3. Material and methods

### 3.1 Surface sediment sampling

Surface sediment samples were recovered from the western Arctic Ocean, including the

Bering Sea, the Bering-Chukchi Shelf and the High Arctic Ocean. 52 surface sediment samples (blue dots in Fig. 1) were collected on the RV *Xuelong* during CHINARE cruises ARC3, ARC4, ARC5 and ARC6 in 2008, 2010, 2012 and 2014, respectively and analysed for biomarkers. In addition, 36 samples obtained during the CHINARE cruises between 2008 and 2014 mostly located in the Chukchi Sea published by Bai et al. (2019) (red dots in Fig. 1) were also used in

this study. The total sample set of 88 surface sediments encompasses a latitudinal range from 54.6°N to 85.4°N. Details of sampling locations are provided in Table S1. Surface sediment sampling was carried out using a box-corer. The uppermost 0-2 cm of sediment were sliced on board then placed in a plastic bag and stored at -20°C until further processing.


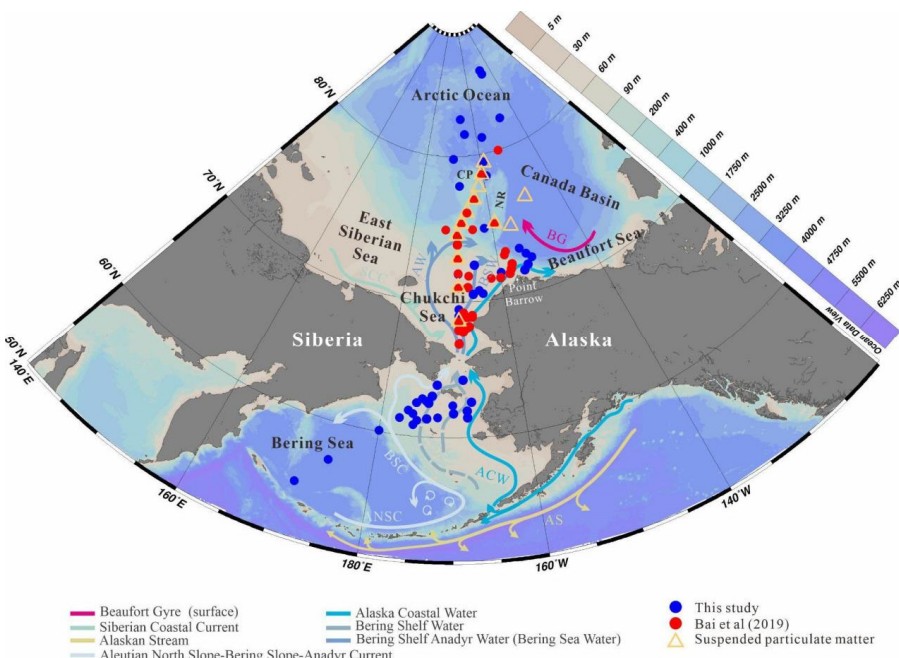

**Figure 1.** Map showing the location of the 88 surface samples in the Bering Sea and western Arctic Ocean, which includes the new data generated in this study (blue dots) and those earlier published by Bai et al. (2019) (red dots). Orange triangles indicate the sites where surface suspended particulate matter was collected (this study). Arrows feature oceanic surface currents (after Grebmeier et al., 2006). AS-Alaskan Stream; ACW-Alaska Coastal Water; ANSC-

Aleutian North Slope Current; BSC-Bering Slope Current; SCC-Siberian Coastal Current; AW-Anadyr Water; BSW-Bering Shelf Water; BG-Beaufort Gyre; CP-Chukchi Plateau; NR-Northwind Ridge. The study area covered the Bering Sea, Chukchi shelf (water depths <140m) and the slope and basin of the western Arctic Ocean (water depths >140m).

*3.2 Suspended particles sampling*

Suspended particulate matter (SPM) was obtained from the filtration of surface seawater at depths ranging from 0 to 5 m during the summer cruise of the RV *Xuelong* in 2014. A Large Volume Water Transfer System (WTS-LV, Mclane) was used to perform *in situ* filtration of seawater (40-60 L) using glass fibre filters (Whatman GF/F, 142 mm diameter) pre-combusted

at 450°C for 4h. After filtration, the particulate-laden filters were stored at -20°C until biomarker analysis. The 13 stations occupied in the Chukchi shelf and Canada basin in summer 2014 (29 July-14 August) are shown in Fig. 1 (orange triangles).

Water samples were also collected for chlorophyll *a* analysis using 1 L Niskin bottles during the same summer 2014 cruise on the RV *Xuelong*. Surface water (0.5-1 L) were first filtered

through 200 μm Nitex filters to remove zooplankton then sequentially filtered through GF/F filters (Whatman GF/F, 25 mm diameter).



### 3.3 Total organic carbon (TOC) and Total nitrogen (TN) sediment analyses

Approximately 1g of freeze-dried and homogenized sediment was acidified with 1M
hydrochloric acid (HCl) and heated in a water bath at 50°C for at least 48 hours to remove
inorganic carbon. Samples were then rinsed with milli-Q grade water until neutral pH was
reached (pH=7) and freeze-dried to remove water (Williford et al., 2007; Su et al. 2023). The
TOC and TN contents of the surface sediments were measured using an element analyser
(FLASH 2000 CHNS-O, Thermo Fisher). BBOT standard of Thermo (carbon% 72.53%;
nitrogen% 6.51%) were used for quality control. BBOT standard was measured every 8 samples
to correct for drift. The standard deviation of the measurements is less than <0.1%.

### 3.4 Sample preparation and biomarker extraction

Wet SPM filters and surface sediments were freeze-dried prior extraction with organic
solvents. Lipids were extracted from freeze-dried SPM and from ca. 1-5 g of homogenised
sediments using a mixture of dichloromethane/methanol (2:1, v/v). Sediment extraction was
performed in a clean glass vial for 10 min in an ultrasonic bath, which was then centrifuged for
2 minutes at 2500 rpm. The supernatant containing the lipids was recovered with a clean glass
pipette and transferred in a pre-combusted 8 mL glass vial. This operation was repeated twice
and the three extracts combined and dried under a gentle nitrogen stream. For filters, each
freeze-dried sample with SPM was cut into small pieces using solvent rinsed scissors and
extracted following the same method as for the sediments. Total lipid extracts were then
separated into n-alkanes, HBIs and sterols by silica gel chromatography using n-hexane, n-
hexane/ethyl acetate (90:10 v/v) and n-hexane/ethyl acetate (70:30 v/v), respectively. About 50
µl BSTFA (bis-trimethylsilyl-trifluoroacetamide) was added to the sterol fractions to convert
them into their corresponding trimethylsilyl ethers by heating at 70 °C for 1 hour to complete
derivatization before analysis by gas chromatography (GC) on an Agilent Technologies 7890
gas chromatograph coupled to a mass spectrometer (MS) Agilent 262 Technologies 5975C inert
XL using a mass selective detector.


### 3.5 Biomarker analysis

$C_{25}$-HBIs ($IP_{25}$, HBI-II, and HBI-III) were analysed by GC/MS. We used a HP-5MS capillary
column (30 m long, 0.25 mm i.d., 0.25 µm film thickness) and an oven temperature program
from 40 °C to 300 °C at a heating rate of 10 °C/min with a 10 min hold time at final temperature.
The operating conditions for the MS were 250 °C for the ion source temperature and 70 eV for
the ionisation energy. HBIs were identified by comparing their GC retention times and mass
spectra. A known amount of 7-hexylnonadecane (*m/z* 266) was added as an internal standard



prior extraction for HBI quantification. Selective ion monitoring (SIM) was performed to detect and quantify $IP_{25}$ (*m/z* 350), HBI-II (*m/z* 348) and HBI-III Z isomers (*m/z* 346).

Sterols and *n*-alkanes were analysed by GC using a Varian 3300 equipped with a septum programmable injector (SPI) and a flame ionisation detector (FID). For *n*-alkanes, the oven temperature was programmed from 80 °C to 300 °C at a hearing rate of 8 °C/min and the final temperature held for 20 min. A 30-m long DB-5MS fused silica capillary column (0.25 mm i.d., 0.25 µm film thickness) was used for both compound classes. For sterols, the GC oven was

programmed from 50°C to 100°C (30°C/min), then from 100°C to 150°C (1.5 °C/min) and to 300°C (3°C/min) held for 20 min. In both cases, a known amount of 5α-cholestane was added to the sample prior GC analysis as an external standard for quantitation. All biomarker sediment concentrations were normalised to TOC.

### 3.6 H-Print index

The HBI based-index, H-Print, was calculated to estimate the relative contribution of pelagic *versus* sympagic sources (Brown et al., 2014b; Koch et al., 2020).

$$H - print\% = \frac{[HBI-III]}{[IP_{25}]+[HBI-II]+[HBI-III]} \times 100 \qquad (1)$$

Low values of H-Print% are indicative of higher relative sympagic production while high

ones reflect prevailing pelagic algae producers. This index was used as an additional information source together with phytosterols to assess phytoplankton production patterns in relation to sea ice conditions.

### 3.7 Chlorophyll a analyses

After collection, filters for chlorophyll *a* (Chl *a*) analysis were extracted with 10 mL of 90% acetone at -20°C in the dark for 24 h and measured using a trilogy laboratory fluorometer (10-AU, Turner Designs), which was calibrated before analysis (Holm-Hansen et al., 1965; Welschmeyer, 1994). The precision of the measurements is 0.02 mg m$^{-3}$.

### 3.8 Sea ice distribution data





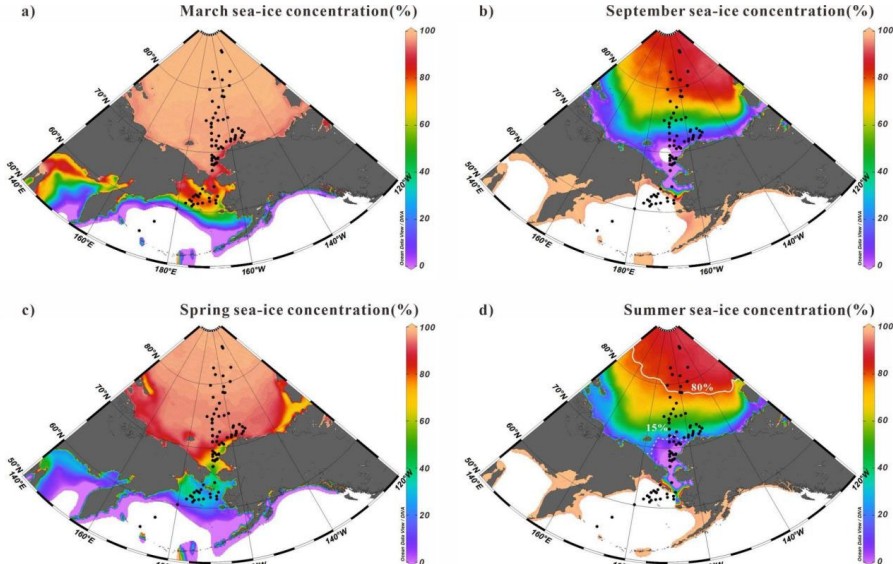

**Figure 2.** Average sea-ice concentrations in (a) March, (b) September,(c) spring (April, May and June) and (d) summer (July, August and September) from 1988 to 2007 (http://nsidc.org). Black dots are surface sediments samples in this study. The dotted and plain white lines represent the 15% and 80% isolines of summer sea ice cover for the period 1988-2007.

Satellite sea ice concentrations in the study region measured from Nimbus-7 SMMR and DMSP SSM/I-SSMIS passive microwave sensors averaged between 1988 and 2007 (Fig. 2a-d). These data were retrieved from the National Snow and Ice Data Center (NSIDC) (Cavalieri et al., 1996, http://nsidc.org). This time interval was selected partly because it has been widely used in the most sea-ice proxy calibration to date (Xiao et al., 2015; Smik et al., 2016; Bai et al., 2019). In addition, Navarro-Rodriguez et al. (2013) demonstrated that a 20-year time-interval satellite time series for mean sea ice concentration was reasonably consistent with sea ice cover datasets in recent decades, regardless of their exact time frame. However, it must be kept in mind that the first 2 cm surface sediments may represent a longer time interval (decades to centuries) than modern satellite data (Polyak et al., 2009; Stein et al., 2010; Pearce et al., 2017). Fig. 2 shows the seasonal distributions of the sea ice extent in spring (April-June) and summer (July-September) as well as in March and September. Also shown, are the 15% and 80% isolines that indicate that the summer Ice-Free Zone (IFZ <15% of sea ice) was on average located as far north as 72°N and that the Marginal Ice Zone (MIZ, sea ice cover from 15% to 80%) reached up to 80°N latitude.

*3.9 Description of ODV*





The distributional maps of TOC, TN and biomarker concentrations were generated with the Ocean Data View software package, version 5.6.5 (ODV, odv.awi.de, Schlitzer, 2023). They were created using the DIVA (Data Interpolating Variational Analysis) display styles gridding for representing sea ice data from NSIDC and with the coloured dots to visualise biomarker data.

## 4. Results

### 4.1 Total organic carbon (TOC) and total nitrogen (TN) in surface sediments

TOC values of surface sediments vary from 0.1 to 2.2% (1.03±0.63%, n=52) with highest values localised in the mid-Chukchi Sea Shelf and Bering Sea, and lowest ones at latitudes > 75°N (Fig. 3a) (Table S1). The Bering Strait exhibits relatively low TOC content. The spatial distribution of TN shows similar patterns with values ranging from 0.01% to 0.29% (0.14±0.07%, n=50) (Fig. 3b), e.g. one order of magnitude lower than TOC, which results in C/N ratios of about 8 to 10 at all sites except for some northernmost ones that have C/N values roughly two times lower (Fig. 3c).

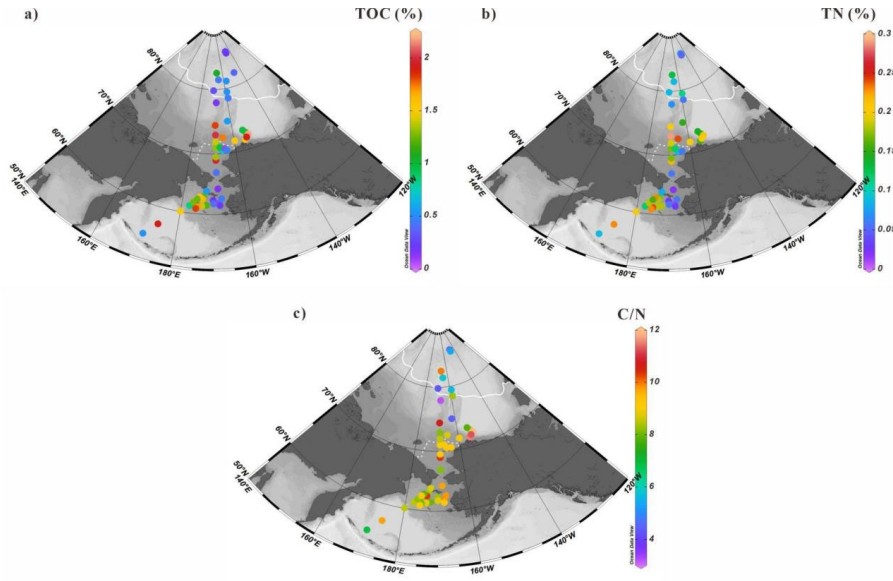

**Figure 3.** Distributions of (a) Total Organic Carbon (TOC) in %, (b) TN (Total Nitrogen) in % and (c) C/N ratios in the surface sediments from the western Arctic Ocean, the Chukchi Sea and the Bering Sea (this study). For an explanation of dotted and plain white lines, see Fig. 2.

### 4.2 C25 high branched isoprenoid (HBI) alkenes in surface sediments

The sea ice biomarker $IP_{25}$ was identified in 47 out of the 52 new surface sediments analysed in this study (Fig. 4a). $IP_{25}$ was absent (or below the detection limit) at locations in the Southern



Bering Sea or at highest latitude sites between 78°N and 82°N. Concentrations were generally high in the northeast Chukchi Sea and Barrow Canyon than in the Chukchi Plateau and low in high latitude areas where sea ice is perennial and predominantly ice-free such as at the Bering Sea slope and shelf ($IP_{25}$: 0.05-1.68 µg g$^{-1}$ TOC, 0.39±0.41µg g$^{-1}$ TOC). HBI-II was also present

in most of the sediment extracts (49 out of 52). Its spatial distribution shared similar features as $IP_{25}$ with higher values matching those of $IP_{25}$ (HBI-II: 0.06-3.02 µg g$^{-1}$ TOC, 0.61±0.63 µg g$^{-1}$ TOC; Fig. 4b) and lowest ones at highest latitudes and in most of the Bering Sea sites.

   HBI-III was present in 45 out of 52 of the sediment extracts. In contrast to $IP_{25}$ and HBI-II, highest abundance (> 0.17µg g$^{-1}$ TOC) were encountered in the northern Bering Sea and at

some sites of the northeast Chukchi Sea where variable sea ice conditions prevailed in summer (Fig. 2d) and were free of ice in September (Fig. 2b), as expected from HBI III producers preferably growing in sea ice edge to ice free waters (0.01-0.94 µg g$^{-1}$ TOC, 0.17±0.22 µg g$^{-1}$ TOC; Fig. 4c). These results are consistent with a pelagic phytoplankton origin for this biomarker (Belt et al., 2015; Bai et al., 2019; Su et al., 2022) and reflected by H-Print%

indicating highest values in the northern Bering Sea gradually decreasing northwards with increasing seasonal sea ice cover, e.g. lowest value at MIZ about 73ºN, slight increase northward from the Chukchi border to high latitudes under perennial sea ice. (Fig. 4d).

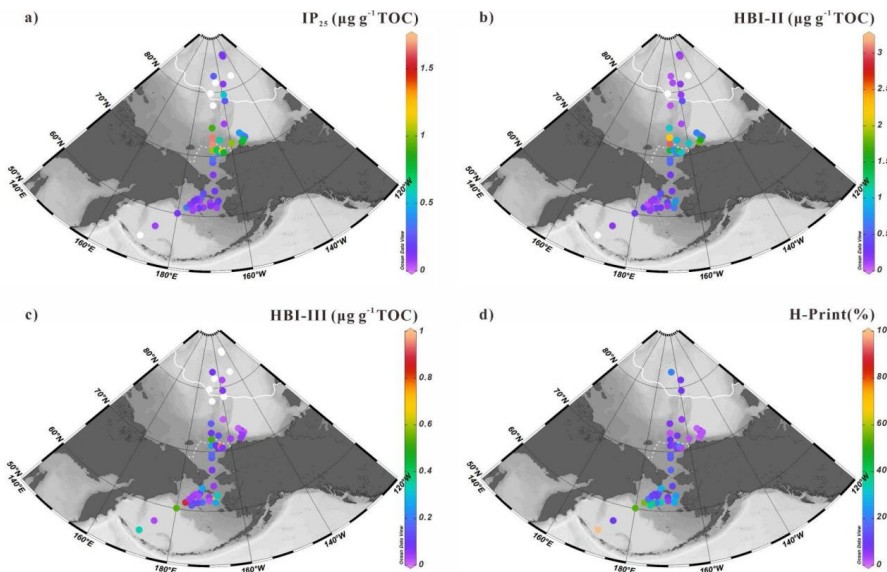

**Figure 4.** Surface sediment concentrations of HBIs normalized to TOC (expressed in µg g$^{-1}$
TOC): (a) $IP_{25}$, (b) HBI-II, (c) HBI-III; (d) values of H-Print % of the western Arctic Ocean, the Chukchi Sea and the Bering Sea (this study). White dots are sites where HBIs were not detected. For an explanation of dotted and plain white lines, see Fig. 2.





### 4.3 Sterols in surface sediments

Fig. 5 shows the spatial distribution of prevailing and commonly used sterols in marine settings to identify phytoplankton communities. They include pelagic phytosterols, e.g. brassicasterol (24-methylcholesta-5,22(E)-dien-3ß-ol) a common sterol find in diatoms, dinosterol (4α-23,24-trimethylcholest-22(E)-en-3ß-ol) mainly produced by dinoflagellates, campesterol (24-methylcholest-5en-3ß-ol) of terrestrial origin and the 24-ethylcholest-5-en-3ß-

ol, which depending on the configuration of C24 that can be of marine (    isomer) or terrestrial (ß isomer) origin (Volkman, 1986; 2003).

Concentrations of brassicasterol range from 0.72  to 57.67 µg g$^{-1}$ TOC (7.92±10.62 µg g$^{-1}$ TOC) with highest values found in the northern Bering Sea (57.67 µg g$^{-1}$ TOC) and southeastern Chukchi Sea shelf (0.72-37.56 µg g$^{-1}$ TOC) (Fig. 5a). Dinosterol concentrations vary in quite a

similar range, from 0.61 to 43.04 µg g$^{-1}$ TOC, around a slightly higher mean value (14.92±11.77 µg g$^{-1}$ TOC). The distribution pattern of dinosterol shows similar features as brassicasterol with high values in the mid- Chukchi Sea shelf and northern Bering Sea but expanding further South (Fig. 5b). Campesterol and 24-ethylcholest-5-en-3ß-ol show high values notably along the coast of Alaska and decrease off-shore across the Bering Strait (campesterol: 0.47-123.41 µg g$^{-1}$ TOC

(30.88±31.36 µg g$^{-1}$ TOC); 24-ethylcholest-5-en-3ß-ol ranges from 0.67 to 152.35 µg g$^{-1}$ TOC (39.52±42.08 µg g$^{-1}$ TOC)) (Fig. 5c, d). These sterols were also both present at low levels at lowest and highest latitudes.

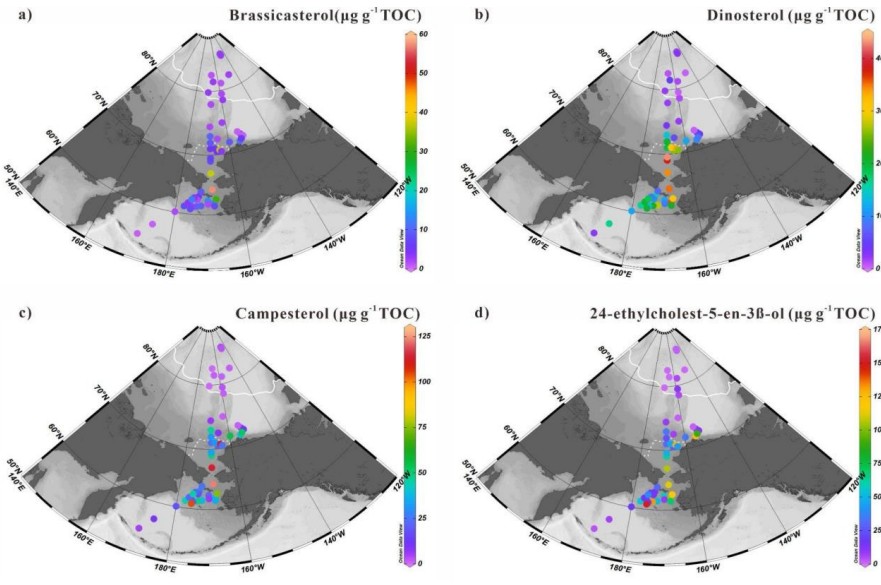

**Figure 5.** Concentrations of sterols expressed in µg g$^{-1}$ TOC: (a) brassicasterol, (b) dinosterol, (c), campesterol and (d) 24-ethylcholest-5-en-3ß-ol in surface sediments from the western





Arctic Ocean, the Chukchi Sea and the Bering Sea (this study). For an explanation of dotted and plain white lines, see Fig. 2.

**4.4 N-alkanes in surface sediments**

The homologous distribution of *n*-alkanes in our sediments display an odd-to-even predominance characteristic of epicuticular waxes produced by higher plants (Yunker et al., 1995). The sum of the concentrations of high molecular weight odd carbon numbered chain *n*-alkanes ($C_{27}$, $C_{29}$ and $C_{31}$) was calculated to assess terrestrial inputs to surface sediments. The

$\sum C_{27}$-$C_{31}$ values shown in Fig. 6 vary from 11.67 to 211.86 µg g$^{-1}$ TOC (84.16±43.44 µg g$^{-1}$ TOC) with higher contents in the southeastern Chukchi Sea shelf across the Bering Strait till the North Bering Sea. Moderate values (>100 µg g$^{-1}$ TOC) were found at some sites of the High Arctic.

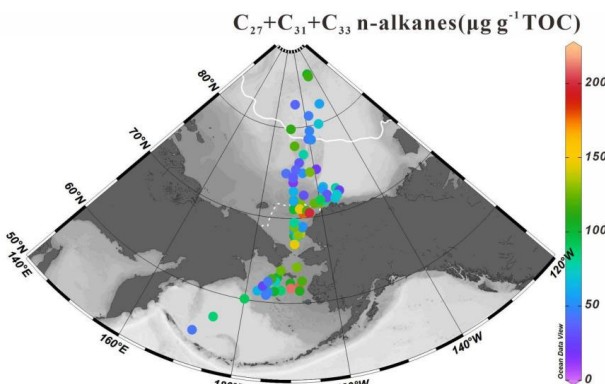

**Figure 6.** Sum of terrigenous long-chain odd numbered $C_{27}$+$C_{29}$+$C_{31}$ n-alkanes (expressed in µg g$^{-1}$ TOC) in surface sediments (*n*=88) from the western Arctic Ocean, the Chukchi Sea and the Bering Sea (this study). For an explanation of dotted and plain white lines, see Fig. 2.

**4.5 Sterols and chlorophyll a in surface suspended particles**

Fig. 7 shows the sterol concentrations measured in surface suspended particles collected at 13 stations located along a latitudinal transect between 68.6°N and 79.4°N (orange triangles in Fig. 1 and Fig. 7f). They include brassicasterol, dinosterol, 24-ethylcholest-5-en-3ß-ol and cholest-5-en-3β-ol (cholesterol). Note that neither HBIs nor campesterol were detected in SPM samples indicating their absence or trace levels, below the detection limit. Concentrations of

brassicasterol ranged from 0.80 to 132 .31 ng L$^{-1}$, 17.11±37.46 ng L$^{-1}$) (Table S3) with highest values found in the mid-Chukchi Sea shelf where chlorophyll *a* also reached highest values and lowest ones North of 75°N (Fig. 7a). Dinosterol concentrations range in a much lower range from 0.51 to 17.11 ng L$^{-1}$ (4.07±5.39 ng L$^{-1}$) and display similar trends as brassicasterol (>10 ng L$^{-1}$, Fig. 7b). Likewise, 24-ethylcholest-5-en-3ß-ol also shows increased concentrations

southwards spanning from 1.88 to 46.04 ng L$^{-1}$ (9.11±12.13 ng L$^{-1}$) (Fig. 7c). Finally,




cholesterol, one of the most abundant sterols in SPM, varied from 1.20 to 117.29 ng L$^{-1}$ (17.02±32.04 ng L$^{-1}$) in a range that is similar to brassicasterol (Fig. 7d). Despite possible algal sources, cholesterol is considered to be mainly reflecting the presence of zooplankton as it converts much of the sterols produced by algae into cholesterol. Chlorophyll *a* in surface waters

show the same South to North latitudinal decrease with concentrations spanning from 0.05 to 5.40 mg/m$^3$ (0.69 ±1.66 mg m$^{-3}$)(Fig. 7e).

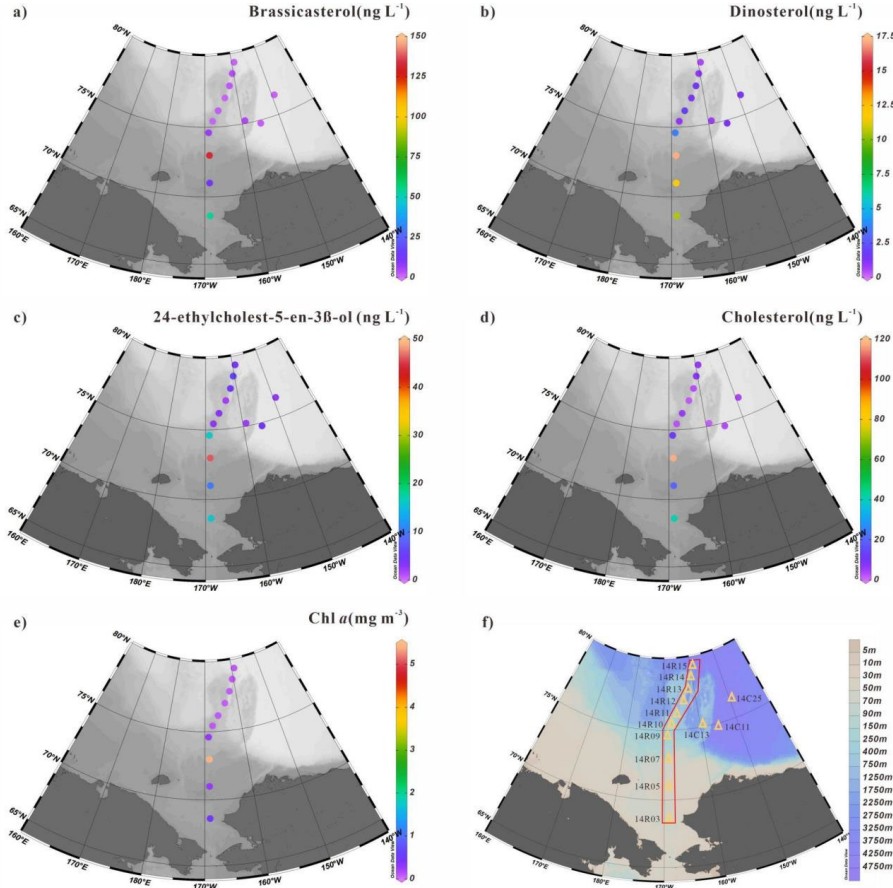

**Figure 7.** Distribution of biomarker sterols (in ng L$^{-1}$) and chlorophyll *a* (in mg m$^{-3}$) in surface suspended particles collected in summer 2014:(a) brassicasterol, (b) dinosterol, (c) 24-

ethylcholest-5-en-3ß-ol, (d) cholesterol, (e) chlorophyll *a*. (f) transect showing the 10 out of 13 stations from the shelf to the basin in the Chukchi Sea where suspended particulate matter was collected.

## 5. Discussion

*5.1 Impact of sea ice on sedimentary HBIs distribution*



In this section, we discuss the spatial distribution of HBIs based on the compilation of the sedimentary data from the CHINARE cruises ARC3, ARC4, ARC5 and ARC6 in the Chukchi Sea and Bering Sea (this study) and those from the CHINARE cruises ARC3, ARC5 and ARC6 published by Bai et al. (2019) obtained following the same analytical procedure (Fig. 8).

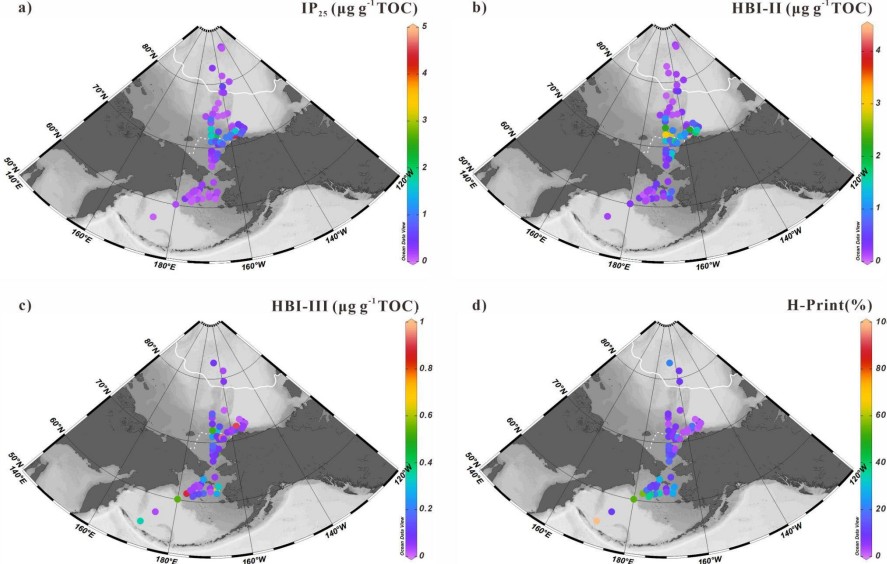


**Figure 8.** Surface sediment concentrations of HBIs normalized to TOC (expressed in µg g$^{-1}$ TOC): (a) IP$_{25}$, (b) HBI-II, (c) HBI-III ; (d) values of H-Print % in the sites (*n*=88) of the western Arctic Ocean, the Chukchi Sea and the Bering Sea. For an explanation of dotted and plain white lines, see Fig. 2.


Sympagic IP$_{25}$ and HBI-II values of the combined data set range from 0.05 to 4.39 µg g$^{-1}$ TOC (0.45±0.61 µg g$^{-1}$ TOC) and from 0.06 to 4.46 µg g$^{-1}$ TOC (0.66±0.75 µg g$^{-1}$ TOC), respectively (Fig. 8a-b, Tables S2). These values are close to those reported by Bai et al. (2019) spanning from 0.09 to 4.39 µg g$^{-1}$ TOC (0.52±0.80 µg g$^{-1}$ TOC) and from 0.12 to 4.46 µg g$^{-1}$ TOC (0.72±0.89 µg g$^{-1}$ TOC) for IP$_{25}$ and HBI-II, respectively. They confirm the extreme values off the Alaskan coast and provide new lower extremes in the Bering Sea slope and shelf, not included in the study of Bai et al. (2019), where ice free conditions prevail in summer (Fig. 8a-b). The strong correlation between sedimentary IP$_{25}$ and HBI-II ($r^2$= 0.87, *p*<0.05, Fig. S1) is in agreement with these two structural homologues having common sources in the Arctic Ocean and northern North Atlantic (Belt, 2018) and with the presence of HBI-II in each all IP$_{25}$-producing species studied by Brown et al. (2014a). Overall, the distribution patterns of sympagic IP$_{25}$ and HBI-II in the augmented dataset confirms the earlier finding of low values at latitudes >73°N due to persisting sea ice in summer and provide new low end-member values South of the Bering Strait essentially free of ice in summer. Enhanced sympagic production is



generally found on the path of the warm polar wards PW and ACW flowing along the coast of Alaska both contributing to ice melting. Comparison with other published data shows that our $IP_{25}$ values are lower than reported in the Chukchi Plateau and basin by Xiao et al. (2015), lying between 0.63 and 8.97 µg g$^{-1}$ TOC, a result that is explained by the generally more northern position of their sites and possibly also different quantification methods. Lowest $IP_{25}$ reported

by Méheust et al. (2013) (0.079 to 0.567µg g$^{-1}$ TOC) in a few (3 out of 11) surface sediments of the Bering Sea are also in agreement with our results and expected from seasonal sea ice in the Bering Sea shelf break, North of the March sea ice edge (20% isoline in Méheust et al., 2013). The northward latitudinal decrease of $IP_{25}$ in surface sediments and elevated values around 70°N were also reported by Koch et al. (2020).

The distribution pattern of pelagic HBI-III is expectedly different from that of sympagic HBIs and exhibit much lower abundances (0.01 to 0.94 µg g$^{-1}$ TOC, 0.16±0.21µg g$^{-1}$ TOC) (Fig. 8c, Table S2). HBI-III is absent or present in trace amounts mostly in ice-covered areas of the High Arctic and highest at summer ice-free sites of the northern Bering Sea (Station 10BB01, 0.89 µg g$^{-1}$ TOC, Table S2). Elevated concentrations occurred in the southeastern Chukchi Sea

shelf along the coast of Alaska, as also found by Koch et al. (2020), and in the Bering Sea. These observations translate into H-Print values indicating that pelagic to sympagic diatom production and export increases from North to South across the Bering Strait and further in the northern Bering Sea (Fig. 8d).

Advective processes linked to the PW flow and their impact on sea ice dynamics in shaping

algal communities have been invoked to explain sympagic to pelagic spatial distribution. The rate of sea ice retreat in the Chukchi Sea is closely connected to heat transport by the PW triggering the onset of sea ice melting in spring/summer. Time series from a year-round mooring deployed in the Bering Strait has shown that annual mean transport volume of PW increased by ~70% between 2001 and 2014 (Woodgate, 2018). Woodgate and Peralta-Ferriz

(2021) further calculated an increase of 0.1 Sv/yr between 1990 and 2019 and a corresponding ~0.1°C/yr warming in spring/summer over this time interval. These authors also outlined that concomitantly the warm period extended from 5.5 to 7 months. This is in contrast with the Bering Sea that does not show any significant long-term reduction since 1850, despite a general warming climate, but pronounced decadal scale variability (Walsh et al., 2017) driven by the

Pacific Decadal Oscillation (PDO), the dominant mode of atmospheric variability in the North Pacific.

Overall, based on HBI sedimentary distribution and their translation into H-Print, we were able to discriminate the three following sub-regions (Fig. 9) : I) the Bering Sea characterized by high variability of pelagic to sympagic production reflecting a wide range of sea ice

conditions with sea ice forming in winter and ice-free conditions in summer (Walsh et al., 2017),





II) the productive MIZ waters of the mid-Chukchi Sea shelf with higher sympagic production and export, and lower spread associated with variable seasonal sea ice conditions, III) the slope and western Arctic Ocean basin characterized by dominating sympagic production and slightly less variability reflecting prevailing high sea ice cover.

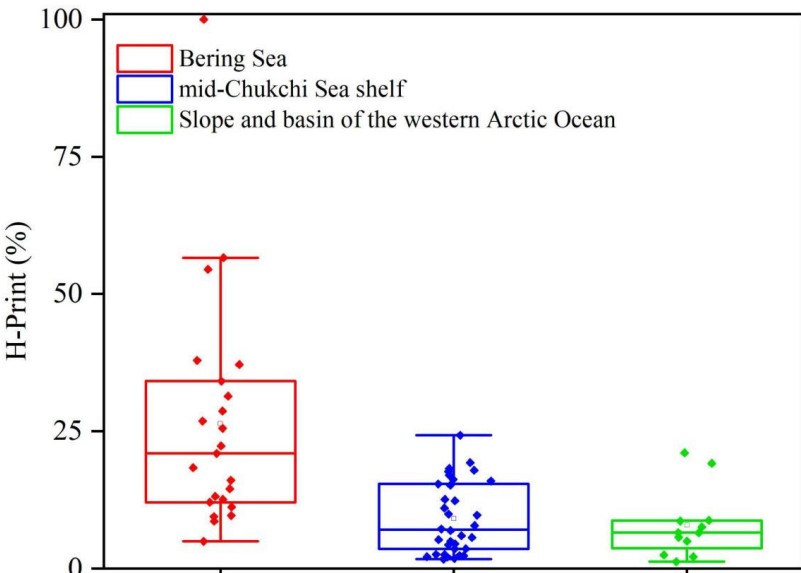


**Figure 9.** Boxplot of H-Print index in surface sediments from the Bering Sea (in red), mid-Chukchi Sea shelf (in blue) and the slope and western Arctic Ocean basin (in green).

### 5.2 Latitudinal variations of sympagic to pelagic phytoplankton production

In order to explore further the relationship between the icescape and sympagic to pelagic

production across the wide range of sea ice situations encountered from the Bering Sea to the High Arctic Ocean, phytosterols (brassicasterol+dinosterol) in surface sediments were investigated and compared with spring and summer sea ice concentrations provided by the NSIDC (Fig. 10a). As shown in Fig. 10b, sympagic production increased from 62°N to 70°N to reach maximum values near the sea ice edge and MIZ, between 70°N and 73°N, and

decreased to low values similar to the IFZ at latitudes >73°N as sea ice became permanent. In contrast, enhanced pelagic phytosterols and HBI-III largely prevailed in open waters of the Bering Sea in spring and summer and decreased relative to sympagic production in the MIZ demonstrating a progressive transition of habitats.





**Figure 10.** Concentrations of (a) spring and summer sea-ice (in %) for the period 1988-2007 retrieved from NSIDC, (b) sympagic HBIs (IP$_{25}$, in µg g$^{-1}$ TOC) ,(c) pelagic HBIs (HBI-III, in µg g$^{-1}$ TOC), (d) terrestrial sterol (campesterol, in µg g$^{-1}$ TOC), (e) pelagic sterol (brassicasterol+dinosterol, in µg g$^{-1}$ TOC) in surface sediments along the latitudinal transect from the Bering Sea to the western Arctic Ocean.



This biogeographic pattern is supported by surface sediment data of Koch et al. (2020) retrieved from the Distribution Biological Observatory (DBO) regions surveyed during several cruises from 2012 to 2017 (Moore and Grebmeier et al., 2018) where abundances of $IP_{25}$ in the northeast Chukchi Sea and Barrow Canyon (71-72.5°N) prevailed over those of lower latitudes (62-68°N). High pelagic HBI-III values in IFZ and MIZ (Fig. 10c) are in agreement with known

producers pelagic diatoms *Rhizosolenia* and *Pleurosigma* living at the sea ice edge and in ice-free waters (Smik et al., 2016; Belt et al., 2017; Belt, 2018). Consistent with our results, enhanced HBI-III has also been reported in surface sediments of the Barents Sea underlying the minimum and maximum April sea ice margin (Belt et al., 2015). Further evidence was provided by sediment trap time-series showing peaking HBI-III export flux at low sea ice/sea ice retreat

at the DM station in the Chukchi Sea slope (Bai et al., 2019; Gal et al., 2022). Dominant pelagic production in the northern Bering Sea is consistent with expanded open water areas and lengthened phytoplankton growing season compared to more northern locations. Sea ice thinning and melting are key factors responsible for increased primary production and northwards expansion of phytoplankton blooms over the recent decades (Renaut et al., 2018).

Physical modelling also evidenced a clear northward shift of phytoplankton blooms from the Bering Sea to the Arctic shelf in the western Arctic Ocean with decreasing sea ice (Jin et al., 2012). Kahru et al. (2016) estimated an increase by 47% of the net primary production across the Arctic Ocean between 1998 and 2015 in the spring and summer. These changes are largely due to enhanced incoming light resulting from earlier ice breakup in summer, later freezing in

winter, increased open water areas and a longer ice-free season boosting the phytoplankton production. In some areas, spring blooms along the sea ice edge contribute more than half of the annual primary production and fall bloom becomes more frequent at latitudes >70°N (Ardyna et al., 2014). Other dynamical factors linked to the decline of sea ice can contribute to increase primary production such as wind-driven mixing and upwelling both leading to the

replenishment of surface waters with nutrients in the Chukchi Sea continental shelf (Pickart et al., 2011; Huntington et al., 2020). However, freshening associated with sea ice melting and subsequent ocean stratification counteract these processes by reducing nutrient availability in surface waters in summer - early fall. Finally, the PW represents a major source of nutrients for phytoplankton growth across the Bering Strait and along the coast of Alaska, which also allows

the intrusion of temperate species in the Arctic Ocean (Ardyna and Arrigo, 2020).

Pelagic phytosterols (e.g. brassicasterol and dinosterol) along our latitudinal gradient share strong resemblance with HBI-III but show higher values relative to HBI-III in the IFZ, e.g. in the southern Chukchi shelf between 68 and 70°N (Fig. 10,c-e). At latitudes >78°N, on the slope and western Arctic basin under perennial sea ice, pelagic phytosterols drop to their lowest



levels as a result of light and nutrient limitation (Gosselin et al., 1997; Fernández-Méndez et al., 2015). This result suggests that diatom production below sea ice is not significant at these extreme latitudes probably because of sea ice thickness preventing light transmission as also revealed by the OC and TN contents of the sediments (Arigo et al., 2012, Coupel et al., 2012).

Interestingly, campesterol displays broadly similar features as phytosterols with higher
values found across the Bering Strait, at the northern Bering Sea and on mid-Chukchi Sea shelf (Fig. 10d). The occurrence of branched and isoprenoid tetraethers in surface sediments in the Bering Sea and inner shelf of the Chukchi Sea (66-73°N) (Park et al., 2014) as well as $\delta^{13}$C values < -25‰ (Naidu et al., 2004; Grebmeier et al., 2006; Ji et al., 2019) further support a higher contribution of land-derived inputs to the sediments that is also reflected by our C/N
values (Fig. 11c, d). Using the BIT index as a proxy of terrestrial carbon, Park et al. (2014) reported low values on the outer shelf of the Chukchi Sea (73-75°N), moderate values offshore the Bering Sea, inner shelf of the Chukchi Sea (66-73°N) and western Arctic Ocean north of 75°N, and highest values in the Yukon and Mackenzie River estuaries. With the loss of continental ice, increasing temperature and precipitation, enhanced biomass production on land
and intensified erosion have resulted in increased river discharge of suspended particulate matter in the coastal Arctic Ocean. Note that higher plant *n*-alkanes (Fig. 6) show no latitudinal trend and unexpectedly high values in the High Arctic that demonstrate that allochthonous material reaches the central Arctic Ocean. The north-west Bering Sea higher value area might be contributed by the Yukon River (Brabets et al., 2000), while high values in the southeastern
Chukchi Sea shelf might be contributed by near-shore sea ice transport, which bring massive black sediments when they formed in near shore close to Barrow (Harper, 1978; Darby et al., 2009).

### 5.3 Comparison of sterols in suspended particles with surface sediments

Biogeochemical data in the water column escapes satellite measurements and are thus very
scarce and critically needed to fully understand the impact of on-going changes on primarily production and the export and sequestration of organic carbon in the Arctic Ocean. As discussed in the previous sections, surface waters properties are very sensitive to environmental changes linked to sea ice that exerts a strong control on primary producers and ultimately food resources (Lannuzel et al., 2020). The amount of exported organic material to the deep ocean depends on
the structure of the phytoplankton community and subsequent higher trophic levels (Leu et al., 2011). The mismatch of spring phytoplankton bloom and zooplankton grazing activity caused by earlier sea ice melting and earlier blooming could lead to lower consumption of phytoplankton by grazing heterotrophs in the water column thereby also affecting export (Hunt et al., 2002; Søreide et al., 2010). In this section we investigate the sterols in suspended particles

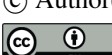
and underlying surface sediments to assess vertical transport pathways, though on a limited

sample set.

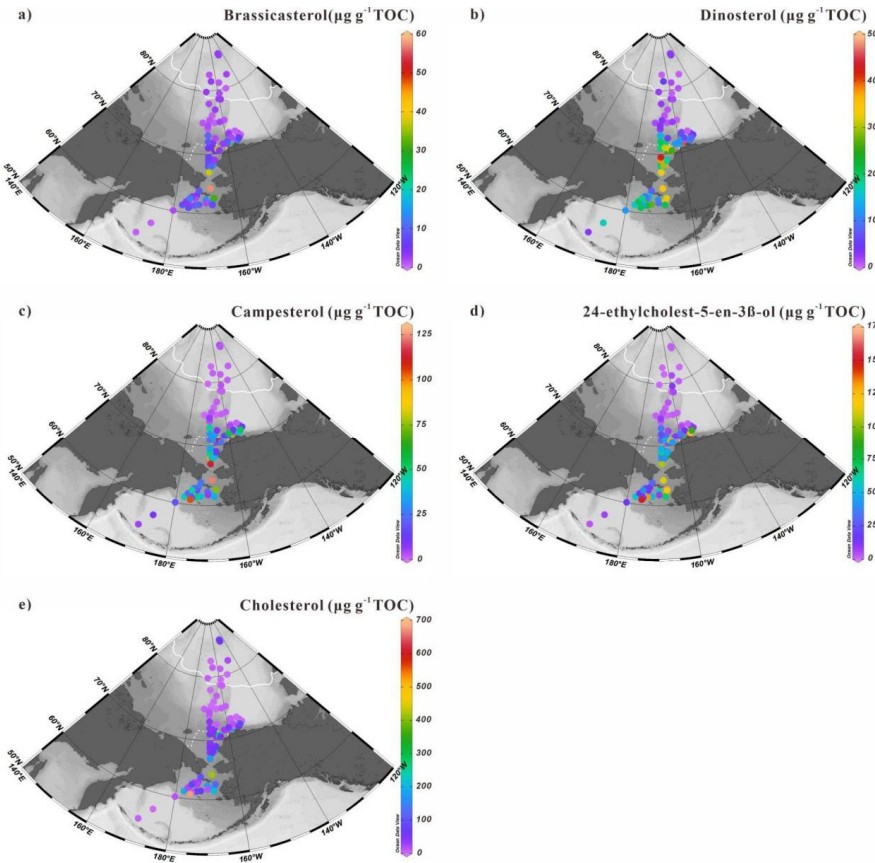

**Figure 11.** Concentrations of sterols normalized to TOC in surface sediments (expressed in µg g$^{-1}$ TOC): (a) brassicasterol and (b) dinosterol; (c) campesterol and (d) 24-ethylcholest-5-en-3ß-ol; (e) cholesterol in the surface sediments (*n*=88) from the western Arctic Ocean, the Chukchi Sea and the Bering Sea. For an explanation of dotted and plain white lines, see Fig. 2.

As earlier outlined, higher contents of brassicasterol and dinosterol are localized in surface

sediments of the Bering Sea shelf and the southeastern Chukchi Sea shelf and decreased sharply

around 73°N to the North with increasing sea ice cover (Fig. 11a,b). Despite the limited SPM

sample set, brassicasterol and dinosterol in SPM reproduce the northward decrease with

increasing sea ice similar as Chl *a* with highest values found in the South of Chukchi Sea shelf

(Fig. 7). The correlation between 24-ethylcholest-5-en-3ß-ol and Chl *a* is significant ($r^2$=0.90,

$p<0.05$) (Fig. S2) and remains high even after the removal of the most extreme value ($r^2$=0.79,

$p<0.01$) (Fig. S3) pointing out a prevailing contribution of the marine origin α isomer 24( )-

ethylcholest-5-en-3ß-ol over the ß isomer, ß-sitosterol, as reported Tolosa et al. (2013) who



demonstrated the prevalence of 24(   )-ethylcholest-5-en-3ß-ol even in coastal waters of the southeast Beaufort Sea. These observations are also in line with minor amounts or below the detection limit of terrestrial biomarkers e.g. campesterol and *n*-alkanes in SPM.

Cholesterol is usually a dominant sterol in zooplankton (Volkman, 1986; 2003) and has often been considered as an indicator of zooplankton activity (Grice et al., 1998), although this compound can also be produced by some algal species. Higher cholesterol abundances were essentially found in northern Bering Sea sediments (Fig. 11e). In suspended particles, linear regression led to correlation coefficients ($r^2$) of 0.99 and 0.79 (both $p>0.05$) for cholesterol against brassicasterol and for cholesterol against dinosterol (Fig. S4). For sediments, we found no significant correlation between cholesterol and brassicasterol or between cholesterol and dinosterol ($r^2=0.37$, $p<0.01$ and $r^2=0.27$, $p<0.01$, respectively; Fig. S5), suggesting other a weak benthos-pelagos coupling. However, phytoplankton and zooplankton living below sea ice and in the subsurface chlorophyll maximum e.g. deeper depth than 5m (our sampling depth) are not accounted in the SPM, while they may represent an important component of primary production and exported material to the sea floor (Arrigo et al., 2012; Coupel et al., 2012). The timing of phytoplankton bloom and zooplankton grazing is another factor that could account for this discrepancy.



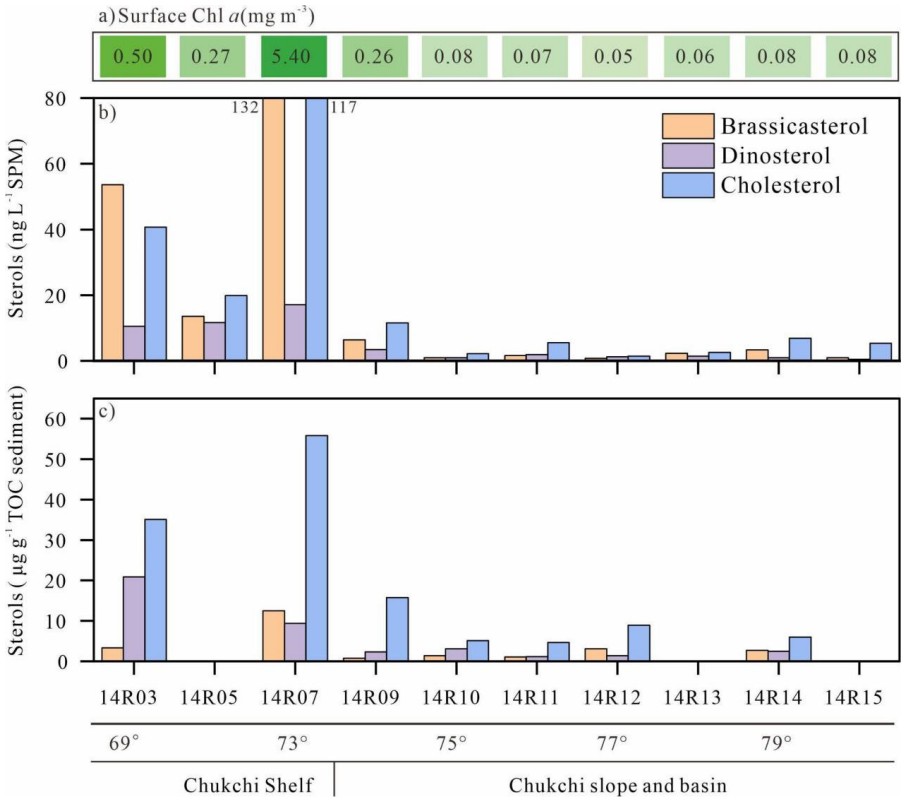

**Figure 12.** Concentrations in surface waters of Chl *a* (a) brassicasterol, dinosterol and cholesterol in SPM (b) and underlying surface sediments (c) along the transect from the shelf to the basin in the Chukchi Sea. For location of the transect see Fig.7. Please note that there is no sediment sample at 14R05, 14R13 and 14R15.

**Table 1** Ratios calculated from individual sterol concentrations in suspended particulate matter (SPM) and sediments from the Chukchi shelf and slope and basin of the western Arctic Ocean.

|  | 14R03 | 14R05 | 14R07 | 14R09 | 14R10 | 14R11 | 14R12 | 14R13 | 14R14 | 14R15 |
|---|---|---|---|---|---|---|---|---|---|---|
| *SPM* | | | | | | | | | | |
| B/C | 1.32 | 0.68 | 1.13 | 0.55 | 0.42 | 0.30 | 0.54 | 0.86 | 0.48 | 0.19 |
| D/C | 0.26 | 0.59 | 0.15 | 0.30 | 0.42 | 0.34 | 0.84 | 0.55 | 0.14 | 0.09 |
| B/D | 5.10 | 1.16 | 7.73 | 1.84 | 1.01 | 0.89 | 0.65 | 1.55 | 3.51 | 1.95 |
| *Sediment* | | | | | | | | | | |
| B/C | 0.10 | — | 0.22 | 0.05 | 0.27 | 0.23 | 0.35 | — | 0.45 | — |
| D/C | 0.60 | — | 0.17 | 0.15 | 0.60 | 0.24 | 0.15 | — | 0.41 | — |
| B/D | 0.16 | — | 1.34 | 0.32 | 0.44 | 0.94 | 2.26 | — | 1.11 | — |

Sterol: (B) brassicasterol (24-methylcholesta-5,22E-dien-3β-ol), (C) cholesterol (cholest-5-en-3β-ol), (D) dinosterol (4α, 23,24R-trimethyl-5α-cholest-22E-en-3β-ol).



Fig. 12 shows the cholesterol and phytosterol concentrations in SPM and underlying surface sediments along a latitudinal transect between 69°N and 79°N latitude. Sterols and Chl *a* both exhibit large amplitude change in surface waters and underlying surface sediments from North to South. Highest brassicasterol and Chl *a* in surface waters of the northern Chukchi shelf at station 14R07 reflect the highly productive waters, underlined earlier in the discussion, with a

major contribution of diatom pigment fucoxanthin (Fig. 12a-b, Table S1). Zhuang et al. (2020) reported high biomass of diatoms (>20 µm) dominating in the subsurface chlorophyll maximum during the same cruise in summer 2014, South and North of the Chukchi shelf. This hotspot of primary production is sustained by continuous nutrient supply by the PW both during the ice melting season and ice free season. The second highest Chl *a* and brassicasterol values of the

transect are found at the southern site 14R03 bathed by low nutrients Alaska Coastal Water that indicate the dominance of small diatoms (< 20 µm) (Zhuang et al., 2020). To the North beyond the shelf, in the Chukchi slope and basin, SPM and sediments display lowest Chl *a* and sterol values but shows that primary production is still significant beyond the continental shelf. Similarly, Gosselin et al. (1997) reported that both phytoplankton and ice algal production were

highest over the Chukchi shelf (70°-72° N) and decreased polewards with increasing sea ice cover and/or the depth of the surface mixed layer, both determining the light and nutrient available to microalgae growth. Arrigo et al. (2012) found massive under-ice phytoplankton bloom over the Chukchi shelf around 73°N supported by light transmission through first-year ice (0.8-1.3 m in thickness) due to thinning sea ice and proliferation of melt ponds. Dominating

brassicasterol over dinosterol in surface SPM in the Chukchi shelf (stations 14R03, 14R05 and 14R07) indicate diatom-rich waters in the MIZ (Table 1). However, underlying sediments evidence limited export of brassicasterol compared to dinosterol suggesting efficient grazing of diatoms and their transfer in the food web. At the southernmost station 14R03, phytoplankton in surface waters was dominated by small diatoms that are more inclined to remain and degrade

in the upper layer, because they can hardly sink if not grazed (Table S1). Another explanation for different vertical transport is that 14R03 experienced ice free and stratified conditions in summer that favour dinoflagellates development late in the season and their more efficient export to surface sediments while diatoms seem to be mostly consumed early in the spring bloom season. High cholesterol abundances in mid-Chukchi Sea shelf SPM and underlying

sediments reflect the consumption by herbivorous zooplankton and export through faecal pellets in the richest phytoplankton waters (Fig. 12b-c). It also underlines the preservation of organic matter in shallow depth sediments of the Chukchi shelf. With the exception of site 14R05, for which sediment data are missing, these finding are illustrated by brassicasterol / cholesterol ratios (B/C) mainly >1 and highly variable in the SPM over the Chukchi shelf, and

B/C<1 in the Chukchi slope and basin (Table 1). Prevailing cholesterol in surface sediments

associated with B/C<1 and D/C<1 further emphasizes the role of zooplankton and grazing pressure on primary producers in the rich waters of the MIZ and likely efficient transfer to higher trophic levels.

## 6. Conclusions

Using the HBI biomarkers and H-Print sea ice index of 88 surface sediments distributed along a latitudinal transect from the Bering Sea till the High Arctic, we were able to distinguish three biogeographic regions: i) the Bering Sea exhibiting a high variability of enhanced pelagic to sympagic production reflecting a wide range of sea ice conditions of the IFZ more strongly influenced by the PDO than global warming, ii) the mid-Chukchi Sea shelf productive waters characterized by higher and less variable sympagic production due to variable sea ice conditions of the MIZ, and  iii) the slope and western Arctic Ocean basin displaying higher sympagic relative to pelagic production constrained by a higher sea ice cover and narrow variability. Our results also underline the role of PW inflow in shaping local environmental changes and primary production in the mid-Chukchi Sea shelf located in the MIZ through heat and nutrient transport.

Our data evidence a conspicuous latitudinal trend in phytoplankton population with increasing sympagic relative to pelagic production northward and highest production and export of both sea ice microalgae and pelagic diatoms in the MIZ in the mid-Chukchi continental shelf. Low terrestrial sterols and n-alkanes in surface sediments underscore a low terrigenous contribution to the sediments with no clear latitudinal trend highlighting different sources and transport pathways than marine organic matter. This is further confirmed by their absence or below detection level in suspended particles between 68.6°N and 79.4°N. Quite unusually, we also found that the 24-ethylcholest-5-en-3ß-ol was essentially represented by the marine    - isomer, the 24(  )-ethylcholest-5-en-3ß-ol, even at most coastal sites.

Comparison of phytosterols and cholesterol fingerprint between SPM and surface sediments suggest efficient consumption of ice microalgae and pelagic diatoms during the spring blooms as compared to dinoflagellates that generally thrives under stratified conditions towards the end of the production season. However, more investigation in the water column and of the sinking particle flux is needed to get a comprehensive picture of the fate and behavior of phyto- and zooplankton. An integrated approach combining trophic biomarkers such as fatty acids and carbon stable isotopes, could serve as a basis to explore further the links between phyto- and zooplankton in this complex environment  and improve our understanding of the fate of sea ice algae and pelagic production and their dynamics across the food web. This knowledge is essential to evaluate the future evolution of marine resources for local populations as well as for assessing the amount of OC transferred and sequestered to the sea floor to also advance our



understanding of carbon cycle feedback mechanisms on climate in this rapidly changing Pacific Arctic system.

**Data availability**

All data that support the findings of this study are included with the article (and supplementary information files).

**Author contributions**

Y. B. and J. C. designed the study. M.-A. S. initiated the writing of the paper with Y. B.  J.
R. for reviewing the paper. V. K. contributed the biomarker analyses (HBIs and sterols) and paper content. H. J. acquired funding from Chinese Arctic and Antarctic Administration and made edits on the final version.

**Competing interest**

The authors declare that they have no known competing financial interests or personal relationships that could have appeared to influence the work reported in this paper.

**Acknowledgements**

We are grateful to the captain, crew members and scientific party of the R/V *Xuelong* for
their help with the retrieval of the surface water suspended particulate matter and surface sediments. We also thank Fanny Kaczmar who helped produce HBI data at LOCEAN and to Dr. Guillaume Massé for supplying chemical standards for HBI quantification. We thank the Centre National de la Recherche Scientifique (CNRS) for salary support of M.-A.S. and V. K. and for biomarker analyses. We thank Zhongqiang Ji, Liang Su and Yang Zhang for TOC and
TN determinations of surface sediments, Dr. Hongliang Li and Yanpei Zhuang for sediment sampling and Qiang Hao for chlorophyll analyses in the surface water. We also thank Dr. Xiaotong Xiao of the Ocean University of China for assistance in retrieving sea ice concentration data. The pigments data used in this work are available in the Data-sharing Platform of Polar Science (https://datacenter.chinare.org.cn/) maintained by the Chinese
National Arctic & Antarctic Data Center (CN-NADC).

**Financial support**

This study was funded by the National Key Research and Development Program of China (Nos. 2019YFE0120900), the National Natural Science Foundation of China (Nos. 41976229,
41941013, 41776205, 42076241 and 42376243), the Scientific Research Funds of the Second Institute of Oceanography, MNR (No. JG2310), the Chinese Polar Environmental





Comprehensive Investigation and Assessment Programs (No. CHINARE 0304), and the project ICAR (Sea Ice melt, Carbon, Acidification and Phytoplankton in the present and past Arctic Ocean) funded by Cai Yuan Pei Program.





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
