# Peer review of "Latitudinal distribution of biomarkers across the western Arctic Ocean and the Bering Sea: an approach to assess sympagic and pelagic algal production"

_Biogeosciences, 2023_

## Author Response (AR1)

**Reply to associate editor** Author **replies are in Blue.**

Associate editor's comment:

Dear Drs Bai and Chen et al.,

Thanks for submitting your manuscript to Biogeosciences and for your responses to the comments received from two reviewers. I largely agree with the assessment of both reviewers and you have responded satisfactorily to most of their suggestions, so I look forward to your revised manuscript that includes the proposed corrections and improvements.

Additionally, I would like you to add/demonstrate more clearly the new scientific contribution you make and therefore the additional value of this new manuscript, especially in comparison with your previous study. I understand the reasons for this new study, but this should be highlighted more clearly in your manuscript, which was a comment that was also raised by reviewer 2.

Previous studies focused on evaluating HBIs as proxies of sea-ice cover (Bai et al., 2019), this one is dealing with assessing sympagic and pelagic production using HBIs and sterols, therefore the purpose is quite different as reflected by the title and subsequently this is definitely a new contribution. This is re-emphasized at the end of the introduction.

Also, reviewer 1 raised concern that the H-print has been calculated incorrectly, which you clarified and explained well in your response, but you have not indicated how you would change this in your revised manuscript. I think you should add some explanation to the manuscript to prevent any confusion to other readers. Similarly, in a few of your responses, I was not sure how (or if) you intend address related comments/suggestions in the revised manuscript or if you simply answered them without any corrections to the manuscript, so I would like to see the explanations in the response file and as well.

There is some misunderstanding here. The calculation of the H-Print index was properly

done. Indeed, H-print never used RF-corrected concentration values but analytical intensities and this is how it has been calculated in this study. Therefore, we were able to compare our results with other studies (Koch et al., 2020, Marine Ecology Progress Series). We explained better how H-Print was originally defined by Brown et al. (2014, Environmental Chemistry Letters) and how it evolved later to be used for sediments or other type of samples (Brown and Belt, 2017; Koch et al., 2020a, b and 2023; Su et al., 2023) that do not necessarily contained the 7 HBIs as in Brown et al. (2014, Environmental Chemistry Letters).

The new version of the rebuttal letter now includes the renumbered lines for the revised sections of the manuscript taking into account addition information on H-Print in the method section added after the letter of the editor.

Thanks for these edits and I look forward to the next version of your manuscript.
Bai et al. have analyzed a new set of sediments (2008, 2009, 2012, 2014) and filtered seawater SPM across the Pacific Arctic region using HBI biomarkers, sterols and *n*-alkanes. The findings of this study are consistent with other similar studies in this particular region and within similar years. These new data points and additional markers do not significantly advance our understanding of HBI synthesis or dynamics in the region, but do validate previously reported observations by Bai (2019) and Koch (2020). However, a notable contribution of this study to the existing body of knowledge is the inclusion of well-defined end members for the H-print index with respect to sea ice conditions, including ice free waters in the Bering Sea near the Aleutians up through perennial ice cover in the Chukchi borderlands and analysing alongside additional sterol biomarkers.

While the results are relatively straightforward, there are a few issues or suggestions to consider further:

In the Introduction, perhaps it is no longer necessary to provide the extensive explanation of HBIs as it is rather well established in the growing body of literature on HBIs (lines 77-91). Perhaps this could be more consider by simply refer to the sea ice HBIs (IP25 and HBI-II) and the pelagic HBI III, and exclude the information about the history of the biomarker (e.g., initially proposed by Belt in 2007, used in paleo studies by Masse).

This section in the introduction has been rephrased and shortened in the revised version (Lines 79-86).

It is unclear why the HBI data needs to be presented twice (once without the Bai et al. 2019 data points and this study). They don't appear to be all that different. I suggest using the one map to show the different data sources (as done in Fig 1) and only report the whole dataset (n=88) in a figure (Fig 8). A comparison of the two could be better

served in a table – and if not that different, perhaps just in the supplemental data?

The reason why we opted for this format is that new data are to be presented in the result section. The discussion section allows to compare with other results and since these new data were acquired following the same procedure it made sense to have figures combining older and new data and used this exercise as a sensitivity test.

Major issue – In reviewing the supplemental data, I fear the H-print has been calculated incorrectly. The "relative abundance" of HBIs is not the value quantified against the standard – which is the absolute abundance. It is the raw data from the SIM chromatograms. It is these relative abundance values that should be used in the H-print calculation. I am happy to connect with the authors to advise on this. I believe this is why the H-print values are much lower than expected in Figs. 4d and 8d for the Bering Sea up through Bering Strait, indicating a dominant sympagic source. Figure 9 will also need to be revised with the new values.

Thanks for your raising this issue. The relative abundances of $IP_{25}$, HBI II and HBI III in the supplemental Table 2, reported as $\mu g\ g^{-1}$ TOC, are based on the area of individual HBIs obtained from SIM by GC-MS relative to the internal standard. They are not corrected by the response factors (RF) as in the original paper of Belt et al. (2012). To clarify the confusion: our data are indeed relative abundances because they are calculated relative to the internal standard, while those corrected with the RF are absolute abundances.

H-Print was originally defined as the ratio of HBIs from sympagic and pelagic diatoms by Brown et al. (2014) and Brown & Belt (2017). We used HBI relative concentrations derive H-Print index as in Koch et al. (2020, 2021) using Eq. (1), where H-Print does not refer to RF-corrected values as in our case (Lines 219-224).

H-Print (%) = [HBI III/ ($IP_{25}$+HBI II+HBI III)] ×100          (1)

We thus compared our data to those from other authors who calculated it the same way.

For Figure 9, there does not appear to be a significant difference between the means of the Chukchi Shelf and slope regions which the authors indicate are distinct.

Indeed, the Chukchi Shelf and slope regions have similar means but different spread (standard deviation) highlighting lower variability in sea ice cover to the North reflecting extended sea ice cover. This is the point we want to make in figure 10.

In this discussion section, I also recommend looking at and comparing with the Koch et al. 2020 paper from Marine Ecology Progress Series which expands the sediment dataset into 2018 and is presented similarly.

In the revised version, this remark has been considered in Section 5.1(Lines 415-416).

I like how the biomarkers are presented with sea ice data in Figure 10.

Sea ice concentration data used in the final version were obtained for the interval 1994-2014 as reviewer 2# suggestion.

Line 461 – this was also reported by Koch et al. 2020 through the sediment trap data linking HBI III with the mentioned genera. Apologies for repeatedly referring to this paper but there is a great deal of overlap.

We added Koch et al. (2020) there as well (Lines 465-466).

Line 552 suggests weak benthic-pelagic coupling based on the sterol rations in SPM versus sediment. I'm not convinced by this and the region is known for having strong B-P coupling. Is the use of sterol ratios in this way based on previous work?

We rephrased this part of the discussion to clarify (Lines 555-557). Differences between SPM and surface sediments involves other mechanisms than vertical transport, such as grazing. The time span of each sample type may also account for the observed differences.

Line 615 in the conclusion states the influence of PDO rather than global warming. This was not tested in this study.

Here, we suggest based on previous discussion (Lines 425-429) that refers to the findings of Wash et al. (2017), that since 1850 despite global warming PDO variability

was dominant in shaping the phytoplankton community in this region of the North Pacific, which is also consistent with the absence of a long-term reduction of sea ice there.

Other editorial suggestions include:

Line 49-50: change to "at the onset of the spring bloom"; remove "still"

Done. See Lines 50-51.

Line 52: change "is" to "are"

Done. See Line 53.

Line 62: change to "mixotrophic"

Done. See Line 63.

Line 78: remove "indeed"

Done.

Line 108: suggest using "community composition" rather than phytoplankton structure

Done. See Line 110.

Line 283: change to "higher" rather than "high"

Corrected. See Line 285.

Line 294: suggest including Koch et al 2020 a & b (PLOS One and MEPS papers)

Done. See Line 297.

Line 310: an issue with the symbol character displaying as a square

Corrected. The symbol should be displayed as α isomer.

Line 515: should be primary production, not primarily

Corrected. See line 521.

Line 552: change to benthic-pelagic

Done. We rephrased this sentence. See Lines 555-557.
Bai et al. present a record of 52 surface sediment samples and 13 suspended particulate matter samples from the upper 5m of the water column from the Arctic Ocean, Chukchi Sea and Bering Strait/Sea. In all samples they have analyzed biomarkers for sea ice as well as phytoplankton productivity and terrigenous input, in order to investigate the spatial distribution of these compounds in surface sediments and waters.

The presented record confirms the distribution of the analyzed compounds as already published findings from this area and working group. Bai et al. add the H-Index as a relatively new method to calculate pelagic to sympagic productivity, which nicely underlines the distribution patterns and the applicability of this index.

I apologize, in case I have misinterpreted anything.

We appreciate the referee review. Please refer to the one to-one response below in blue for specific modifications and clarifications.

As mentioned before, this manuscript presents a valuable dataset, however so far, the discussion focusses mainly on the comparison with Bai et al. 2019, which raises the question of the scientific contribution of this manuscript. The presented data set however is valuable and I am sure that the scientific contribution could be higher when the discussion and data comparison is extended.

I do not see a reason why the Bai et al (2019) data is plotted for comparison but the Koch et al. (2020), Xiao et al. (2015) and Méheust et al. (2015) is not plotted if applicable. This would allow for a much better discussion of the distribution in comparison to sea ice extend and productivity. In this regard, the discussion could benefit from a discussion regarding the different seasonality's of algae bloom/biomarker production, as the study region reached from high arctic to the middle latitudes, where light availability and seasonality differ greatly.

See answer to REV#1 above. We have added more than 50 new data that essentially

provide a better representation of the low end-member of HBIs (ice free). The range of $IP_{25}$ concentrations in our surface sediments indicate comparable concentrations as in previous surface studies in the pan-Arctic (Méheust et al., 2015; Xiao et al., 2015; Koch et al., 2020).

Rather than seasonality *stricto sensus*, it is the sea ice cover that drives the relative abundances of sympagic/pelagic productions across the range of latitudes in our data set, which is the focus of this study in the context of global warming (as reflected by the title).

In this regard, it may be useful to also calculate the PIP25 indices for sea ice reconstructions. I feel the study region is not yet studies in detail since the implementation of HBI III as phytoplankton marker in the PIP25 calculation.

$PIP_{25}$ assessment was the focus of Bai et al., (2019). Here, the purpose is to assess sympagic and pelagic algal production and the potential of H-Print and sterols to do so.

I feel that terrigenous sterol distribution is not given enough attention. Especially in regard to the influence of terrigenous input from sea ice and the influence on phytoplankton productivity.

We added a discussion on this issue in the revised manuscript (See section 5.2 Lines 511-513). Changes in sea ice conditions influence terrestrial inputs to the sediments through transport pathways such as sea ice drifting (Eicken et al., 2005; Darby et al., 2009). Regarding the role of terrestrial inputs on primary production, this is still debated including in region like the Mediterranean Sea surrounded by land masses. Nutrient availability originating from aerosols on primary production and notably the concept of bioavailability has been a research topic for many decades. This is different from coastal waters where riverine inputs of nutrients, essentially in inorganic forms, are known for having a direct impact on primary production.

**Specific Comments**
**Abstract**

The Abstract includes a lot of detail, describing the distribution of biomarkers. I feel this could be shortened and less descriptive.

After a careful reading of the abstract we found that all sentences are needed to provide an understandable description of this work.

**L35** I apologize, if this comment is based on my lack in biological knowledge: What is the connection to pelagic sterols and Chl a with diatoms. Aren't sterols and Chl a produced by a wide variety of phytoplankton species?

Diatoms are often associated with brassicasterol (24-methylcholesta-5,22E-dien-3β-ol). Based on the similarity of this sterol and Chl *a,* we suggest that primary production is dominated by diatoms. This is further supported by studies such as Zhuang et al. (2020) who pointed out that spatial variability of Chl *a* point to a high phytoplankton biomass in the south Bering shelf (SBS), south Chukchi shelf (SCS), and North Chukchi shelf (NCS). They also demonstrated that small diatoms dominate the phytoplankton community in the SBS (> 60%), whereas large diatoms prevail in the SCS and NCS (> 90%). Koch et al. (2020) also reported that pelagic diatoms were largely responsible for the Chl *a* distribution in the northeast Chukchi Sea region from August to October 2015, based on taxonomic determination.

[Figure]

The above figure from Zhuang et al. (2020) shows (a) the relative contribution of different algal classes (large diatoms, small diatoms, and other groups) and (b) group

composition of phytoplankton 0.7–20 μm in size to total chlorophyll a (Chl *a*) concentration calculated by CHEMTAX analyses in subsurface chlorophyll maxima layers in the Bering–Chukchi shelf.

*Zhuang, Y., Jin, H., Chen, J., Ren, J., Zhang, Y., Lan, M., Zhang, T., He, J., and Tian, J.: Phytoplankton community structure at subsurface chlorophyll maxima on the western Arctic shelf: patterns, causes, and ecological importance, Journal of Geophysical Research: Biogeosciences, 125, e2019JG005570, 2020.*

**Introduction**

**L49** "annual bloom period" – To my understanding, there are several blooms (spring, summer, autumn), and this is, also in the Arctic Ocean, until September (sea ice minimum)

As the reviewer 1# suggestion, we change "annual" to "at the onset of the spring bloom".

**L51** increases **in** later summer/early autumn

Done.

**L77** exchange diagnostic with geochemical

Done.

**L88** This is the only time you use a hyphen for sea ice. Make sure you stay consistent.

We are very grateful to Reviewer #2 for a thorough reading of the manuscript. We have carefully examined the manuscript to correct and harmonize.

**Material and methods**

**L179** prior **to** extraction

Done.

**L179** delete "with organic solvents"

Done.

**L207** heating instead of hearing

Done.

**L240** I strongly disagree with the arguments given by the authors using the sea ice satellite record from 1988 – 2007. Yes, Navarro-Rodriguez et al., (2014) could show that surface records are not explicitly representing a specific time frame. But I would strongly advice against using the old satellite dataset, especially as this study often discussed the most recent changes in the region and the resulting changes in sea ice, meltwater, river run off etc., and the most dramatic sea ice changes occurred after 2007. Why not choose a 20-year time frame going back from the year of sampling (1994-2014)?

In the revised version, we have used satellite sea ice concentration data from 1994-2014 to generate average sea-ice distributions for spring (April, May and June) and summer (July, August and September). Note that it does not change much the picture and does not have implication on the discussion.

**L247** A appreciate that the age uncertainty of surface sediments is mentioned here. I see that it is often not applicable to confirm the age of surface sediments - However, this study would greatly benefit from it as it will strongly differ within the presented record, which makes me doubt the comparability of the individual stations.

This has been the approach taken by paleoceanography (not even using box-corers). Surface sediment samples integrate the upper few centimeters of sedimentation, and thus represent from several years to century. This is not specific to this study.

We are fully aware of the limitation of this approach and added a sentence in the revised manuscript (more details see section 3.8, Lines 252-254). Nevertheless, distinct latitudinal variations and trends in sea ice biomarkers as well as phytoplankton productivity are consistent with similar studies in this particular region (Koch et al., 2020a, b).

**Results**

**L307** common sterol **found** in diatoms

Done.

**L307** A discussion on the uncertainties of the production of phytoplankton sterols is missing. The producers of brassicasterol and dinosterol are not as straight forwards as presented here and there are more recent publications on this than Volkman (1986), and Volkman (2003).

We agree and re-emphasized the uncertainties of the production of phytoplankton sterols and updated the more recent references (Lines 310-316).

*Volkman, J.K.: Sterols in Microalgae. In: Borowitzka, M.A., Beardall, J., Raven, J.A. (Eds.), The Physiology of Microalgae. Springer International Publishing, Cham,pp. 485–505, 2016.*

**L310** there is a problem with special characters, I only see a square throughout the manuscript

Corrected. The symbol should be displayed as α isomer.

**Discussion**

**L400** This combination of these datasets in a figure would allow a much deeper discussion of the region.

As explained in general comments, there are methodological differences because of RF and the intent is to discuss the trends and patterns within a coherent database to assess sympagic *vs* pelagic productions.

**L414**ff Here, for example, recent changes in the study area are discussed in detail, this is why I see the need to include a more modern sea ice satellite dataset – and also an age control...

Limitation linked to the age of surface sediments is indeed an issue, we have updated a more modern sea ice satellite dataset (1994-2014). Here, we added a comparison with

the result from the surface sediments in Koch et al. (2020b) in the revised manuscript (see section 5.1 Lines 415-416).

*Koch, C. W., Cooper, L. W., Grebmeier, J. M., Frey, K., and Brown, T. A.: Ice algae resource utilization by benthic macro-and megafaunal communities on the Pacific Arctic shelf determined through lipid biomarker analysis, Marine Ecology Progress Series, 651, 23-43, 2020b.*

**L459** The producers of HBI III are not completely confirmed. Please be more cautious here.

Thank you for noting. The sentence has been rephrased (more details see section 5.2 Lines 462-466).

**L469** "recent decades" – do you have proven that your sediments are representing these? Why are you excluding the recent decades

This has been changed by "last decades" in the revised manuscripts (Line 474). However, we are referring here to the paper of Renaut et al. (2018).